# Distribution of Signal Peptides in Microvesicles from Activated Macrophage Cells

**DOI:** 10.3390/ijms241512131

**Published:** 2023-07-28

**Authors:** Kenji Ono, Junpei Sato, Hiromi Suzuki, Makoto Sawada

**Affiliations:** 1Department of Brain Function, Division of Stress Adaptation and Protection, Research Institute of Environmental Medicine, Nagoya University, Nagoya 464-8601, Aichi, Japanhiromi_s@riem.nagoya-u.ac.jp (H.S.); msawada@riem.nagoya-u.ac.jp (M.S.); 2Department of Molecular Pharmacokinetics, Nagoya University Graduate School of Medicine, Nagoya 464-8601, Aichi, Japan

**Keywords:** signal peptides, extracellular vesicles, calmodulin, intercellular communication, macrophages

## Abstract

Extracellular vesicles, such as microvesicles (LEV) and exosomes (SEV), play an important role in intercellular signaling by encapsulating functional molecules and delivering them to specific cells. Recent studies showed that signal peptides (SPs), which are derived from sequences at the N-terminal of newly synthesized proteins, exhibited biological activity in the extracellular fluid. We previously reported that SPs were secreted into the extracellular fluid via SEV; however, it remains unclear whether the release of SPs occurs via LEV. In the present study, we demonstrated that SP fragments from human placental secreted alkaline phosphatase (SEAP) were present in LEV as well as SEV released from RAW-Blue cells, which stably express an NF-κB-inducible SEAP reporter. When RAW-Blue cells were treated with LPS at 0–10,000 ng/mL, SEAP SP fragments per particle were more abundant in LEV than in SEV, with fragments in LEV and SEV reaching a maximum at 1000 and 100 ng/mL, respectively. The content of SEAP SP fragments in LEV from IFNγ-stimulated RAW-Blue cells was higher than those from TNFα-stimulated cells, whereas that in SEV from TNFα-stimulated RAW-Blue cells was higher than those from IFNγ−stimulated cells. Moreover, the content of SEAP SP fragments in LEV and SEV decreased in the presence of W13, a calmodulin inhibitor. Collectively, these results indicate that the transportation of SP fragments to extracellular vesicles was changed by cellular activation, and calmodulin was involved in their transportation to LEV and SEV.

## 1. Introduction

Extracellular vesicles, which contain functional molecules such as miRNAs, mRNAs, and proteins and are released from multiple types of cells, play important roles in intercellular communication between released cells and received cells [1,2]. Extracellular vesicles are expected to be applied to the search for novel biomarkers and drug delivery systems because they deliver degradable and unstable biomolecules to specific cells [3,4]. Extracellular vesicles are classified into the following types based on differences in vesicle sizes and release mechanisms: apoptotic bodies, microvesicles (LEV), and exosomes (SEV). Apoptotic bodies are vesicles larger than 1000 nm in diameter that are released during cell apoptosis and contain many fragmented nuclei and organelles. On the other hand, LEV and SEV are released from living cells. LEV are 100–1000 nm in size and bud from the plasma membrane, while SEV are 30–200 nm in size. The membrane surface of endosomes formed by endocytosis is bowed inward, and intraluminal vesicles are released into the extracellular fluid as SEV. Although LEV and SEV are formed by different processes and have different contents, both contain proteins, mRNAs, and miRNAs as functional biomolecules and deliver them to specific cells, thereby contributing to intercellular communication [5,6]. In a pharmacokinetics study on SEV in mice, their systemic distribution was reported in the kidney, liver, lung, and spleen, whereas LEV predominantly accumulated in the liver [7]. In addition, although there are various mechanisms for the cellular uptake of extracellular vesicles, the most common is endocytosis, whereby extracellular vesicles are engulfed by recipient cells [8]. Another mechanism for LEV uptake is fusion, whereby LEV fuses with the membranes of the recipient cell and releases their contents into the cell. Monocyte-derived LEV fuse with platelets expressing P-selectin, increasing the procoagulability of platelets [9]. Furthermore, macropinocytosis is important for SEV uptake [10]. Therefore, LEV and SEV are considered to play different roles in vivo.

Signal peptides (SPs) have recently been attracting attention as functional biomolecules. They are derived from sequences at the N-terminal tips of newly synthesized proteins that are important for targeting the endoplasmic reticulum. SPs are removed from synthesized proteins on the endoplasmic reticulum by signal peptidase and further cleaved into two fragments by SP peptidases. Although cleaved fragments were previously considered to be degraded intracellularly, some SPs and their fragments have since been detected in extracellular fluid [11,12] and identified as regulators of several cell functions [13,14,15,16,17]. Although the detailed mechanism by which SP fragments are released into the extracellular fluid remains unclear, we recently reported that they were released into the extracellular fluid via SEV [18,19]. We also demonstrated that calmodulin (CaM) bound to SP fragments and played important roles in the transportation of SPs to SEV [19,20]. Since SEV is crucial for intercellular communication, SP fragments in SEV may contribute to intercellular communication as biomolecules that regulate cellular functions. On the other hand, it remains unclear whether SP fragments are released into the extracellular fluid via LEV, another type of extracellular vesicle.

Macrophages are actively involved in intercellular signaling through extracellular vesicles. SEV released from the mouse macrophage cell line RAW264.7 was previously shown to affect the proliferation and differentiation of mesenchymal stem cells [21], and macrophage-derived LEV induced the differentiation of naïve monocytes [22]. Since macrophages are also cells that produce bioactive SPs, such as CCL22 SPs [13], SPs from macrophages may contribute to intercellular signaling via extracellular vesicles. Furthermore, macrophages responded to lipopolysaccharide (LPS), an outer membrane component of Gram-negative bacteria, and secreted many cytokines. Activated macrophages arise in response to interferon-gamma (IFNγ), which may be produced during an adaptive immune response by T helper 1 cells or CD8^+^ cells or during an innate immune response by natural killer cells, and tumor necrosis factor-alpha (TNFα), which is produced by antigen-presenting cells [23]. Macrophages play distinct roles in response to different activators, such as LPS, TNFα, and IFNγ.

In the present study, we demonstrated that the N-terminal fragment of human placental secreted alkaline phosphatase (SEAP) SP was present in LEV as well as SEV and secreted into the extracellular fluid using RAW-Blue cells that stably express an NF-κB/AP-1-inducible SEAP reporter. Distinct activation conditions resulted in distribution differences between LEV and SEV. We also found that CaM was involved in the transportation of SPs into LEV and SEV.

## 2. Results

To confirm the expression of the SEAP reporter, including SP, in RAW-Blue cells by LPS, SEAP activity was measured with conditioned medium (Figure 1A). SEAP activity from RAW-Blue cells increased in a concentration-dependent manner when RAW-Blue cells were treated with 0–10,000 ng/mL of LPS. To examine the properties of LEV and SEV released from LPS-treated RAW-Blue cells, the average size and number of released LEV and SEV were assessed using the NTA method. No significant differences were observed in average sizes with and without the LPS treatment (Figure 1B). The number of LEVs released gradually decreased, reaching a minimum of 100 ng/mL. On the other hand, no significant changes were detected in the number of SEVs released (Figure 1C). When the amount of protein per 10^10^ particles was measured, LEV was approximately five-fold higher than SEV (Figure 1D). In addition, the amount of protein per 10^10^ particles in LEV increased in the presence of LPS, whereas no change was observed in SEV. CD63, a tetraspanin, is a well-known marker against SEV and multivesicular endosomes [2], whereas ADP-ribosylation factor 6 (ARF6) is useful as a marker of LEV from macrophages [24]. To confirm that extracellular vesicles were properly isolated, markers for extracellular vesicles were examined by Western blotting (Figure 1E). ARF6 was detected in the LEV fraction and CD63 in the fraction, indicating precise isolation. We previously reported that the size of SEAP SP fragments in SEV was *m*/*z* (mass to charge ratio) 1277, measured by MALDI-TOF-MS [19]. Peptide fractions from extracellular vesicles were analyzed by MALDI-TOF-MS, and the *m*/*z* 1277 peptide was detected in both LEV and SEV from LPS-treated RAW-Blue cells (Figure 1F). In addition, the MS/MS analysis identified a peak at *m*/*z* 1277 ± 6 as the N-terminal fragment (1–11) of SEAP SP (MLLLLLLLGLR), which contained an MLL-amino acid sequence, as shown in our previous study [19] (Figure 1G). When signal intensity at *m*/*z* 1277 was measured from LEV and SEV in the presence of LPS at 0–10,000 ng/mL and the signal intensity per particle was calculated as a ratio, the maximum value was 1000 ng/mL for LEV and 100 ng/mL for SEV (Figure 1H). Moreover, the signal intensity of SP per particle was higher in LEV than in SEV. These results indicate that SPs were present not only in SEV but also in LEV, and that the amount of SP per particle was higher in LEV than in SEV for SEAP SPs derived from RAW-Blue cells. Furthermore, the distribution of SPs to each extracellular vesicle may change due to differences in the activation state induced by the LPS stimulation because a 10-fold difference in the concentration of LPS has been shown to induce different responses in macrophages.

To clarify whether different activation states affected the distribution of SP fragments, SP fragments in extracellular vesicles from RAW-Blue cells were examined in the presence or absence of TNFα or IFNγ (Figure 2). When RAW-Blue cells were treated with TNFα at 20 ng/mL or IFNγ at 20 ng/mL for 72 h, SEAP activity was measured with conditioned medium. SEAP activity from RAW-Blue cells was increased by the TNFα or IFNγ treatment, but to a lesser extent by the LPS treatment at 100 ng/mL (Figure 2A). This result indicated that the TNFα or IFNγ treatment as well as the LPS treatment activated NF-κB in RAW-Blue cells. To confirm that different activation states were induced in RAW-Blue cells by the three treatment types, mRNA expression in RAW-Blue cells was examined. CCL3 mRNA levels markedly increased in RAW-Blue cells in the presence of LPS and moderately increased in the presence of TNFα. CXCL10 mRNA levels were markedly increased in the presence of IFNγ. Furthermore, iNOS mRNA levels markedly increased in the presence of LPS and IFNγ. These results indicated that different activation states were induced in RAW-Blue cells in the presence of LPS, TNFα, or IFNγ. To examine the properties of LEV and SEV released from TNFα- or IFNγ-treated RAW-Blue cells, the average size and number of released LEV and SEV were measured by NTA. No significant differences were observed in average sizes with and without the TNFα or IFNγ treatment (Figure 2C). No significant changes were observed in the number of extracellular vesicles in the presence of TNFα, whereas it decreased in the presence of IFNγ (Figure 2D). To investigate whether SEAP SP fragments were present in peptide fractions from extracellular vesicles, the peak at *m*/*z* 1277 was detected by MALDI-TOF-MS (Figure 2E). The ratio of SEAP SP fragments per particle in LEV was higher in the presence of IFNγ than in the presence of TNFα (*p* = 0.0037). On the other hand, the ratio of SEAP SP fragments per particle in SEV was higher in the presence of TNFα than in the presence of IFNγ (*p* = 0.0458). These results indicate that the transportation of SP fragments to extracellular vesicles was changed in a manner that depended on the activation state of RAW-Blue cells.

Since CaM plays an important role in the transport of SP fragments into SEV [20], we investigated whether it contributed to the transportation of SP fragments to LEV as well as SEV. One function of CaM is to regulate the cell cycle, and W13 has been shown to reversibly delay the transition to the S phase in CHO-K1 cells [25]. To confirm that W13 inhibited CaM functions in RAW-Blue cells, the number and viability of cells were assessed in the presence or absence of LPS and/or W13 (Figure 3A–D). When RAW-Blue cells were treated with LPS, proliferation stopped and cell morphology changed to the ameboid form. On the other hand, when cells were treated with W13, a CaM inhibitor, no significant changes were observed in their morphology (Figure 3A). In addition, the number of cells was significantly lower in the presence of W13 as well as LPS (Figure 3B). However, no significant differences were noted in cell viability (Figure 3C). Moreover, the expression of cleaved caspase-3, which indicates the induction of apoptosis, was not detected in the presence or absence of LPS and/or W13 by Western blotting (Figure 3D). These results indicate that W13 functionally inhibited the proliferation of RAW-Blue cells, suggesting that W13 interfered with CaM functions in RAW-Blue cells. To clarify whether W13 affected the translation of SEAP, activity was measured in the presence or absence of LPS and/or W13 (Figure 3E). Although the number of cells decreased, no significant difference was noted in SEAP activity in the presence of W13. To examine the properties of LEV and SEV released from RAW-Blue cells in the presence or absence of LPS and/or W13, the average size and number of released LEV and SEV were assessed by NTA. No significant differences were observed in their average sizes in the presence or absence of LPS and/or W13 (Figure 4A). The number of released LEV decreased in the presence of LPS or LPS and W13, while the number of released SEV decreased in the presence of LPS and W13 (Figure 4B). When the signal intensity per particle at *m*/*z* 1277 was measured in peptide fractions from LEV and SEV, it significantly decreased in the presence of W13 (Figure 4C). These results indicate that CaM contributed to the transportation of SP fragments to LEV as well as SEV.

The abundance of CD63, ARF6, and CaM in RAW-Blue cells was examined in the presence or absence of LPS and/or W13 (Figure 5A). Although no significant changes were detected in the abundance of ARF6 and CaM, that of CD63 increased in the presence of LPS. The abundance of ARF6 and CaM was also examined in extracellular vesicles (Figure 5B). CD63 was enriched in SEV and ARF6 in LEV. The abundance of CD63 in SEV from LPS-treated RAW-Blue cells slightly increased, while that in SEV from W13-treated or W13- and LPS-treated RAW-Blue cells slightly decreased. The abundance of ARF6 in LEV from LPS-treated and W13-treated RAW-Blue cells increased, while that in SEV from LPS- and W13-treated RAW-Blue cells additively decreased. GAPDH was enriched in LEV but not in SEV, and its abundance in LEV from W13-treated or LPS-treated RAW-Blue cells slightly increased, while that in LEV from LPS- and W13-treated RAW-Blue cells was decreased. CaM was rich in LEV and SEV. No significant changes were observed in the abundance of CaM in LEV from W13-treated RAW-Blue cells, while that in LEV from LPS-treated and W13-treated RAW-Blue cells increased. Moreover, no significant changes were detected in the abundance of CaM in SEV from W13-treated RAW-Blue cells, while that in SEV from LPS-treated RAW-Blue cells and from LPS- and W13-treated RAW-Blue cells increased and decreased, respectively. An immunocytochemical analysis was performed to clarify the distribution of CD63, ARF6, and CaM in RAW-Blue cells (Figure 5C,D). Regarding CD63 and CaM, CD63 accumulated in the presence of LPS as well as W13. In addition, CD63 partly colocalized with CaM in the presence of W13 or LPS. Regarding CaM and ARF6, ARF6 was dispersed in the cytoplasm in the absence of LPS and W13 but partly accumulated in the presence of LPS. The accumulation of ARF6 did not appear to colocalize with that of CaM. These results indicate that the LPS and W13 treatments affected the abundance and distribution of CD63, ARF6, and CaM.

## 3. Discussion

We herein demonstrated for the first time that SP fragments were encapsulated in both LEV and SEV, but SEV is consistent with previous findings [18,19,20]. The SP fragment detected in LEV from RAW-Blue cells was the N-terminal fragment of SEAP SP, as we previously reported in HEK-Blue cells [19]. LEV differs from SEV in its origin and functions [5,6]. SEV released from macrophages has been shown to play a role in the differentiation of mesenchymal stem cells [21], and macrophage-derived LEV has induced the differentiation of naïve monocytes [22]. Proinflammatory macrophage-derived LEV exhibit tumor tropism dependent on the CCL2/CCR2 signaling axis and promote drug delivery via SNARE-mediated membrane fusion in contrast to SEV [24]. In addition, although the infection of macrophages with *M. tuberculosis* induced the release of LEV and SEV containing MHC class II, naïve T cells were stimulated by the endogenously processed *M. tuberculosis* antigen presented by LEV but not SEV from *M. tuberculosis*-infected macrophages [26]. SPs in LEV and SEV may play distinct roles in recipient cells because both have different target cell types and uptake mechanisms. Therefore, further studies are warranted on the release of SPs from not only SEV, but also LEV into the extracellular fluid via extracellular vesicles.

The distribution of SPs to LEV and SEV was affected by changes in the concentration of LPS and different treatments by TNFα and IFNγ (Figure 1H and Figure 2E). The abundance of SEAP SP fragments per particle was higher in LEV than in SEV. SEAP activity increased in a concentration-dependent manner with the LPS treatment and also increased with the TNFα and IFNγ treatments. These results indicated that SEAP SPs were also produced along with the synthesis of the SEAP protein in cells; however, a positive relationship was not observed between SEAP activity in cells and the abundance of SEAP SPs in extracellular vesicles. These results indicate that for SEAP SPs in extracellular vesicles, intracellularly produced SEAP SPs are not transported to extracellular vesicles in a concentration-dependent manner; their transportation to LEV and SEV changes depending on differences in the cell activation state. When the abundance of ARF6 in LEV and SEV was examined, it was detected in LEV but not in SEV in the presence or absence of LPS and/or W13 (Figure 1E and Figure 5B). On the other hand, when the abundance of CD63 in LEV and SEV was assessed, it was detected in SEV but not in LEV in the presence or absence of LPS and/or W13. Since ARF6 is a marker of LEV from macrophages [24] and CD63 is a marker of SEV [2], changes in the abundance of SEAP SPs in extracellular vesicles do not necessarily reflect the contamination of different extracellular vesicles. Therefore, there appear to be mechanisms that regulate the transport of SP fragments into extracellular vesicles in a manner that depends on the activation state, which will be examined in future studies. In addition, macrophages respond to multiple activators such as LPS, TNFα, and IFNγ in infectious diseases. As macrophages are actively involved in intercellular signaling through extracellular vesicles and the transportation of SPs into LEV and SEV is dependent on the activation state, SPs in extracellular vesicles might contribute to intercellular communications in an infectious state.

The abundance of SEAP SP fragments in LEV and SEV released from RAW-Blue cells in the presence of W13 was reduced, indicating the involvement of CaM in the release of SPs not only in SEV but also in LEV into the extracellular fluid. We previously reported that CaM bound to SPs in the cytoplasm and were released into the extracellular fluid via SEV, and also that the binding of CaM to SPs was inhibited in the presence of W13 using transformed HEK293 cells [19,20]. Therefore, W13 was assumed to have inhibited the binding of CaM and SEAP SPs in RAW-Blue cells, resulting in a reduction in the transport of SPs to LEV and SEV.

The mechanisms by which SP fragments on the cytoplasmic side are transported to each extracellular vesicle after binding to CaM have not yet been elucidated. CaM is an intracellular calcium-binding protein that is involved in multiple biological processes, such as immune responses, metabolism, and higher functions [27]. CaM responds to various signaling pathways and moves through intracellular organelles [27,28]. Although the transportation of SPs toward SEV and LEV also changed in the present study depending on the activation state of cells, SPs may be transported by the intracellular translocation of CaM in response to intracellular signaling pathways. Although CaM is one of the component proteins of extracellular vesicles [29,30], its role in extracellular vesicles remains unclear. The present results also suggest that CaM is an important molecule for transporting SP to extracellular vesicles.

Endosomal sorting complexes required for transport (ESCRT)-dependent and (ESCRT)-independent pathways are involved in the formation of SEV [31,32]. In the ESCRT-dependent pathway, the ESCRT complex induces the formation of intraluminal membrane vesicles by budding and separating vesicles from the inner membrane surface of multivesicular endosomes [33]. On the other hand, tetraspanins and ceramides have been reported to induce exosome budding in the ESCRT-independent pathway [34,35]. Previous studies reported that CaM and ESCRT components co-localized and accumulated in mold cells in the presence of Ca^2+^ and that peripherin/rds, a tetraspanin present in photoreceptor cells, bound to CaM [36,37]. Therefore, CaM bound to SPs may interact with ESCRT proteins and tetraspanins, such as CD63, and become encapsulated in SEV. LEV are small, plasma membrane-derived vesicles that are released into the extracellular environment by the outward budding and fission of the plasma membrane and present surface membrane proteins, such as integrins and selectins, on the vesicle membrane [38,39]. In addition, ARF6, a marker of LEV from macrophages, is required for growth factor- and Rac-mediated membrane ruffling in macrophages and redistributes from the interior of the cell to the plasma membrane [40]. Since some of these proteins interact with CaM during the signaling pathway [41,42,43], CaM bound to SPs that migrated to the membrane vicinity by the activated state of the cell may be encapsulated in LEV. Therefore, the underlying mechanisms need to be examined in more detail in further research.

## 4. Materials and Methods

### 4.1. Cells

RAW-Blue cells (InvivoGen, San Diego, CA, USA) are murine RAW264.7 macrophages that stably express an NF-κB/AP-1-inducible secreted embryonic alkaline phosphatase (SEAP) reporter gene. One million RAW-Blue cells were plated on a 100-mm Falcon cell culture dish (Corning Inc., Corning, NY, USA) and cultured in Dulbecco’s Modified Eagle’s Medium (DMEM) (Sigma-Aldrich, St. Louis, MO, USA) supplemented with 10% fetal bovine serum, from which extracellular vesicles were removed by centrifugation at 110,000× *g* for 24 h, and Penicillin-Streptomycin (Thermo Fisher Scientific, Waltham, MA, USA) in the presence or absence of 0–10,000 ng/mL LPS, 20 ng/mL TNFα, 20 ng/mL IFNγ (R&D Systems, Inc., Minneapolis, MN, USA), or 10 μM N-(4-aminobutyl)-5-chloronaphthalene-2-sulfonamide hydrochloride (W13) (Tokyo Chemical Industry Co., Ltd., Tokyo, Japan) at 37 °C for 72 h in 5% CO_2_/95% humidified air. Photographs of cells were taken using a microscope equipped with a DP73 camera (Olympus, Tokyo, Japan). Conditioned medium and cells were collected, and the number and viability of cells were assessed by the trypan blue dye exclusion assay using the Countess II FL Automated Cell Counter (Thermo Fisher Scientific). Collected cells were washed twice with phosphate-buffered saline (PBS) and stored as cell pellets at −80 °C.

### 4.2. Measurement of SEAP Activity

SEAP activity in conditioned medium was measured using QUANTI-Blue Solution (InvivoGen). In brief, 20 μL of conditioned medium was mixed with 180 μL of QUANTI-Blue Solution and incubated at 37 °C for 1 h in a 96-well plate. Absorbance at 620 nm was measured using a microplate reader.

### 4.3. Isolation of Extracellular Vesicles from Conditioned Medium

Conditioned medium was centrifuged at 300× *g* at 4 °C for 5 min to remove live cells, and the supernatant was centrifuged at 2000× *g* for 20 min to remove apoptotic vesicles. LEV were prepared by centrifugation at 10,000× *g* for 60 min. SEV were prepared from the supernatant by centrifugation at 110,000× *g* for 70 min and re-suspended in 20 μL of PBS per culture dish. The purity of each fraction was confirmed by Western blotting.

### 4.4. Nanoparticle Tracking Analysis (NTA)

The number and average size of extracellular vesicles were measured using the NanoSight NS300 (Malvern Panalytical Ltd., Malvern, UK). Extracellular vesicles were diluted at 1:100 in degassed water to a final volume of 600 μL and applied through a syringe for measurement. The camera level was increased until all particles were distinctly visible without exceeding a particle signal saturation >20% (level 14–15). Automatic settings for the maximum jump distance and blur settings were utilized. The detection threshold was 5. In each measurement, five 60-s videos were captured under the following conditions: cell temperature, 25 °C; syringe pump speed, 100 (instrument-specific unit); camera, sCMOS; laser, 488 nm blue. After capturing, the number and size of extracellular vesicles were analyzed using NTA 3.2 software build 3.2.16.

### 4.5. MALDI-TOF-MS

Extracellular vesicles (10 μL each), such as LEV and SEV, were dissolved in 90 μL of 8 M urea. Peptides were concentrated from the solution with GL-Tip SDB and GC columns (GL Sciences Inc., Tokyo, Japan) and eluted with 50 μL of 80% acetonitrile supplemented with 0.1% trifluoroacetate. The peptide solution was mixed at 1:1 with 10 mg/mL 4-CHCA in 50% acetonitrile supplemented with 0.1% trifluoroacetate, and 1 μL of the mixture was plated on MTP384 target plate ground steel (Bruker, Billerica, MA, USA). After the plate was air-dried, peptides were measured using ultrafleXtreme MALDI-TOF MS (Bruker). In the MS/MS analysis, amino acid sequences were elucidated within an error of 0.7 Da.

### 4.6. Real-Time PCR

Total RNA extraction and cDNA synthesis were performed as previously described [44]. cDNA was amplified and analyzed with Power SYBR Green PCR Master Mix (Thermo Fisher Scientific) and QuantStudio 12K Flex (Thermo Fisher Scientific) using primer pairs specific to CCL3 (sense primer: ATG AAG GTC TCC ACC ACT GC; antisense primer: GAT GAA TTG GCG TGG AAT CT), CXCL10 (sense primer: TCA CTC CCC TTT ACC CAG TG; antisense primer: TGC TTC GGC AGT TAC TTT TG), inducible nitric oxide synthase (iNOS; sense primer: CTG TGA CAC ACA GCG CTA CA; antisense primer: TGG TCA CAT TCT GCT TCT GG), or glyceraldehyde-3-phosphate dehydrogenase (GAPDH; sense primer: TGC ACC ACC AAC TGC TTA G; antisense: GAT GCA GGG ATG ATG TTC) for 40 cycles (95 °C for 15 s, 60 °C for 60 s). mRNA expression was analyzed by the Delta-Delta Ct method [45], and the ratio of CCL3, CXCL10, or iNOS to GAPDH was calculated.

### 4.7. Western Blotting

Cells and LEV and SEV from RAW-Blue cells were lysed in RIPA buffer (20 mM Tris-HCl, 150 mM NaCl, 1 mM Na_2_EDTA, 1 mM EGTA, 1% NP-40, 1% sodium deoxycholate, 2.5 mM sodium pyrophosphate, 1 mM β-glycerophosphate, 1 mM Na_3_VO_4_, and 1 µg/mL leupeptin, pH 7.5) (Cell Signaling Technology, Danvers, MA, USA) by sonication in iced water. The BCA Protein Assay (Thermo Fisher Scientific) was performed to measure protein concentrations. Homogenates from cells were mixed with loading buffer with or without dithiothreitol (Cell Signaling Technology) to 1 mg/mL. Homogenates from LEV and SEV were mixed with loading buffer with or without dithiothreitol to 0.1 mg/mL. The mixture was boiled for 5 min, and 10 μL of the mixture was separated by SDS-PAGE on e-PAGELs (Atto, Tokyo, Japan) or Choju Gel (Oriental Instruments, Kanagawa, Japan). Separated proteins were then transferred to polyvinylidene difluoride membranes on an iBlot2 Gel Transfer Device (Thermo Fisher Scientific) and blocked in Tris-buffered saline (TBS) supplemented with 5% non-fat dry milk (Cell Signaling Technology) and 0.1% Tween 20 at room temperature for 1 h. Membranes were incubated in the presence of primary antibodies diluted in Can Get Signal Immunoreaction Enhancer Solution 1 (Toyobo, Osaka, Japan) at a 1:6000 dilution at 4 °C overnight. Primary antibodies were directed against cleaved caspase-3 (Cell Signaling Technology), caspase-3 (GeneTex, Irvine, CA, USA), ADP-ribosylation factor 6 (ARF6) (ProteinTech, Rosemont, IL, USA), CD63 (MBL; Medical and Biological Laboratories, Tokyo, Japan), CaM (Abcam, Cambridge, UK), and GAPDH (Fujifilm Wako Pure Chemical Corporation, Osaka, Japan). Membranes were washed with TBS supplemented with 0.1% Tween 20 three times and incubated in Can Get Signal Immunoreaction Enhancer Solution 2 (Toyobo) containing horse radish peroxidase (HRP)-conjugated secondary antibodies (anti-rabbit IgG, anti-rat IgG, and anti-mouse IgG, Cell Signaling Technologies) at a 1:12,000 dilution at room temperature for 1 h. Proteins were visualized by chemiluminescence using Clarity Western ECL Substrate (Bio-Rad, Hercules, CA, USA) and Light Capture II (Atto).

### 4.8. Immunocytochemical Staining

A total of 5 × 10^4^ RAW-Blue cells were plated on gelatin-coated coverslips and cultured in DMEM supplemented with 10% fetal bovine serum, from which extracellular vesicles were removed by centrifugation at 110,000× *g* for 24 h, and Penicillin-Streptomycin (Thermo Fisher Scientific) in the presence or absence of 100 ng/mL LPS and/or 10 μM W13 at 37 °C for 72 h in 5% CO_2_/95% humidified air. Cells were fixed with 4% paraformaldehyde (Fujifilm Wako Pure Chemical Corporation) at room temperature for 10 min and washed with PBS three times. Cells were blocked in PBS supplemented with 1% bovine serum albumin (BSA), 10% normal goat serum (NGS), 0.1% Triton-X, and 0.05% sodium azide at room temperature for 30 min. Cells were incubated with primary antibodies diluted in PBS supplemented with 1% BSA, 10% NGS, 0.1% Triton-X, and 0.05% sodium azide at a 1:250 dilution at room temperature for 1 h. Primary antibodies were directed against CD63 (MBL), CaM (Abcam), CaM (Thermo Fisher Scientific), and ARF6 (ProteinTech). Cells were then washed with PBS three times and incubated in PBS supplemented with 1% BSA, 10% NGS, 0.1% Triton-X, and 0.05% sodium azide containing Alexa488-conjugated anti-rat IgG (Thermo Fisher Scientific) at a 1:500 dilution or Alexa488-conjugated anti-mouse IgG (Thermo Fisher Scientific) and Alexa546-conjugated anti-rabbit IgG (Thermo Fisher Scientific) at a 1:500 dilution, and 5 μg/mL Hoechst33342 (Sigma-Aldrich) at room temperature for 30 min. Cells were washed with PBS three times and mounted on glass slides with ProLong Diamond mounting medium (Thermo Fisher Scientific). Images were taken using a BZ-710 fluorescent microscope equipped with a confocal system (Keyence, Osaka, Japan).

### 4.9. Statistical Analysis

Statistical analyses were performed using a two-tailed *t*-test or one-way ANOVA with Tukey’s post hoc tests. Differences were considered significant when the *p*-value was less than 0.05.

## 5. Conclusions

We herein demonstrated that the release of a SP fragment into the extracellular fluid occurs not only via SEV but also via LEV and that its distribution varies depending on the cell activation state. In addition, CaM played an important role in the transport of SPs to both extracellular vesicles. Since SPs in extracellular vesicles have physiological functions and appear to be crucial for intercellular signaling via their delivery to specific cells, an analysis of the functions of SPs in extracellular vesicles will help to elucidate previously unknown biological phenomena and molecular mechanisms. Furthermore, an analysis of SPs in extracellular vesicles may contribute to the creation of biomarkers for various diseases and the development of new disease therapies in the future.

## Figures and Tables

**Figure 1 ijms-24-12131-f001:**
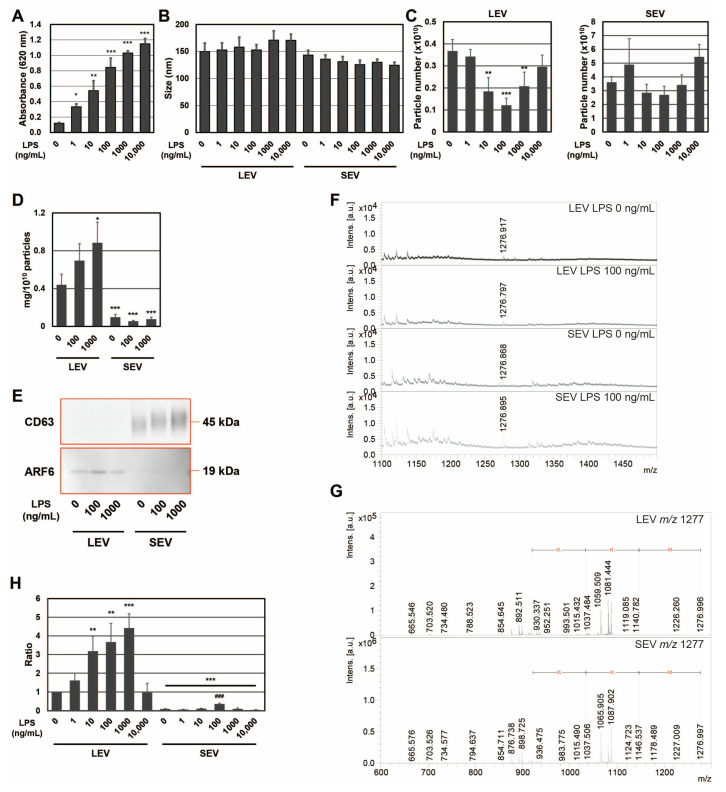
SEAP SPs in extracellular vesicles from RAW-Blue cells in the presence of LPS. (**A**) The SEAP activity of RAW-Blue cells in the presence of LPS at 0–10,000 ng/mL. * *p* < 0.05, ** *p* < 0.01, *** *p* < 0.001 vs. the absence of LPS, *n* = 3. (**B**) The average size of extracellular vesicles from RAW-Blue cells, *n* = 3. (**C**) The particle number of extracellular vesicles from RAW-Blue cells. ** *p* < 0.01, *** *p* < 0.001 vs. the absence of LPS, *n* = 4. (**D**) The amount of protein per 10^10^ particles in LEV and SEV in the presence of LPS at 0–1000 ng/mL. * *p* < 0.05, *** *p* < 0.001 vs. LEV from RAW-Blue cells in the absence of LPS, *n* = 3. (**E**) Western blot analysis of extracellular vesicle-related proteins in extracellular vesicles from RAW-Blue cells in the presence of LPS at 0–1000 ng/mL. (**F**) The peptide solution extracted from extracellular vesicles of RAW-Blue cells at 0 or 100 ng/mL LPS was analyzed by MALDI-TOF-MS. (**G**) MS/MS analysis of peaks at *m*/*z* 1277 ± 6. (**H**) The ratio of the peak intensity per particle at *m*/*z* 1277 was measured by MALDI-TOF-MS. ** *p* < 0.01, *** *p* < 0.001 vs. LEV from RAW-Blue cells in the absence of LPS, ^###^
*p* <0.001 vs. SEV from RAW-Blue cells in the absence of LPS, *n* = 3.

**Figure 2 ijms-24-12131-f002:**
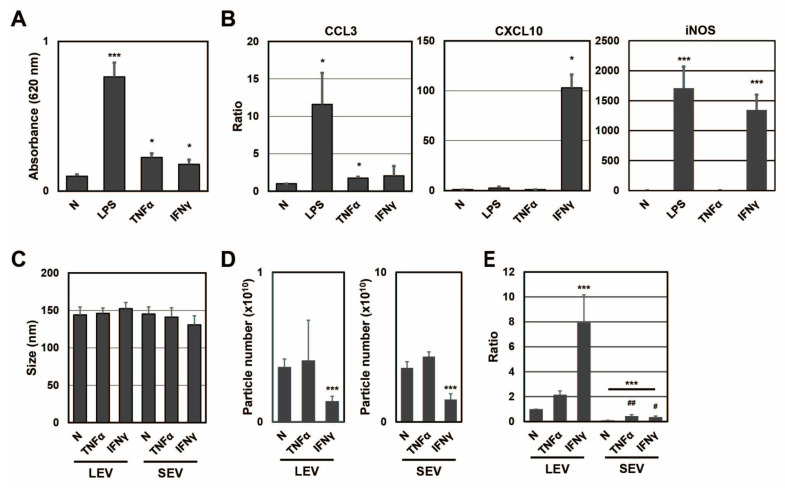
SEAP SPs in extracellular vesicles from RAW-Blue cells in the presence of TNFα and IFNγ. (**A**) The SEAP activity of RAW-Blue cells in the presence of LPS (100 ng/mL), TNFα (20 ng/mL), or IFNγ (20 ng/mL). * *p* < 0.05, *** *p* <0.001 vs. the absence of LPS, TNFα, and IFNγ (N), *n* = 3. (**B**) mRNA expression of CCL3, CXCL10, and iNOS in RAW-Blue cells in the presence of LPS, TNFα, or IFNγ. * *p* < 0.05, *** *p* < 0.001 vs. N, *n* = 3. (**C**) The average size of extracellular vesicles from RAW-Blue cells in the presence of TNFα or IFNγ, *n* = 3. (**D**) The particle number of extracellular vesicles from RAW-Blue cells in the presence of TNFα or IFNγ. *** *p* < 0.001 vs. N, *n* = 4. (**E**) The ratio of the peak intensity per particle at *m*/*z* 1277 was measured by MALDI-TOF-MS. *** *p* < 0.001 vs. LEV N, ^#^
*p* < 0.05, ^##^
*p* < 0.01 vs. SEV N, *n* = 3.

**Figure 3 ijms-24-12131-f003:**
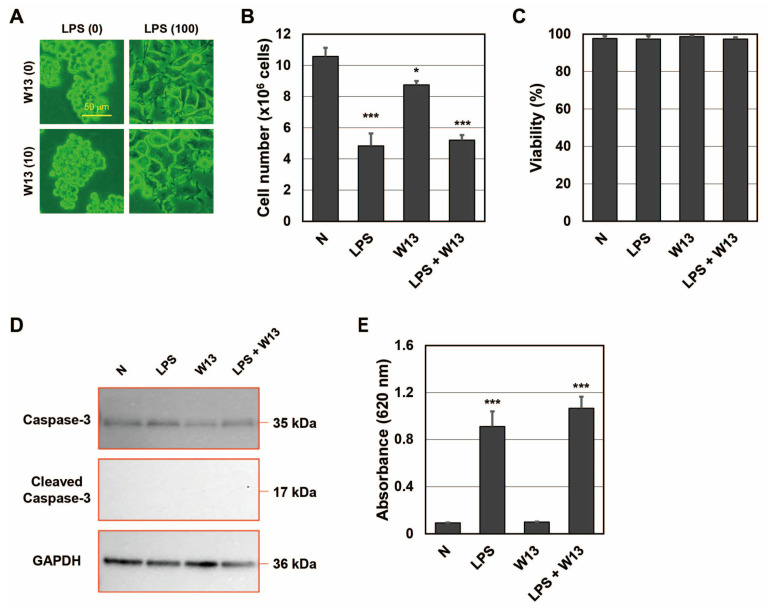
RAW-Blue cells in the presence of LPS and/or W13. (**A**) Representative images of RAW-Blue cells in the presence of 100 ng/mL LPS and/or 10 μM W13. (**B**) The number of RAW-Blue cells in the presence of LPS and/or W13. * *p* < 0.05, *** *p* < 0.001 vs. the absence of LPS and W13 (N), *n* = 3. (**C**) The viability of RAW-Blue cells in the presence of LPS and/or W13, *n* = 3. (**D**) Western blot analysis of apoptosis-related proteins in RAW-Blue cells in the presence of LPS and/or W13. (**E**) The SEAP activity of RAW-Blue cells in the presence of LPS and/or W13. *** *p* < 0.001 vs. N, *n* = 3.

**Figure 4 ijms-24-12131-f004:**
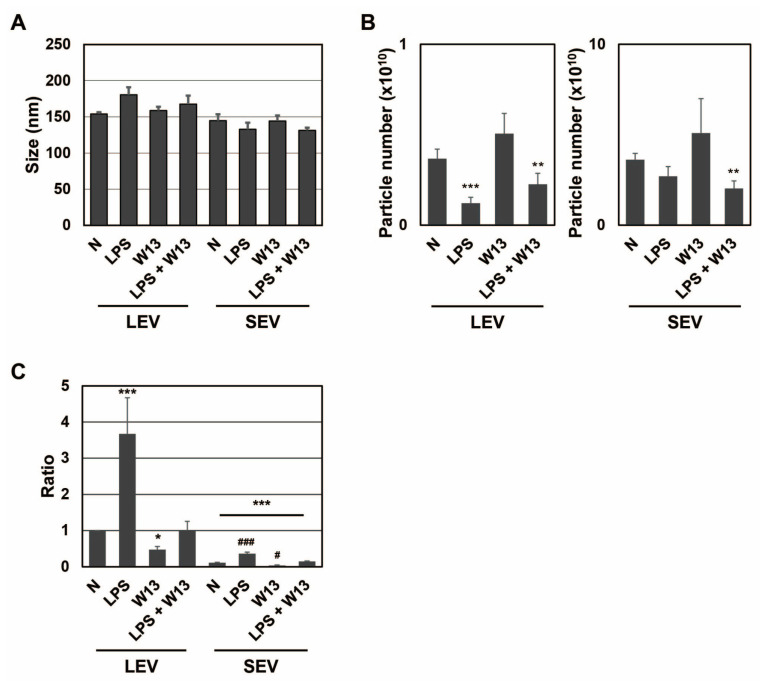
SEAP SPs in extracellular vesicles from RAW-Blue cells in the presence of LPS and W13. (**A**) The average size of extracellular vesicles from RAW-Blue cells in the presence of 100 ng/mL LPS and/or 10 μM W13, *n* = 3. (**B**) The particle number of extracellular vesicles from RAW-Blue cells in the presence of LPS and/or W13. ** *p* < 0.01, *** *p* < 0.001 vs. the absence of LPS and W13 (N), *n* = 4. (**C**) The ratio of the peak intensity per particle at *m*/*z* 1277 was measured by MALDI-TOF-MS. * *p* < 0.05, *** *p* < 0.001 vs. LEV N, ^#^
*p* < 0.05, ^###^
*p* < 0.001, *n* = 4.

**Figure 5 ijms-24-12131-f005:**
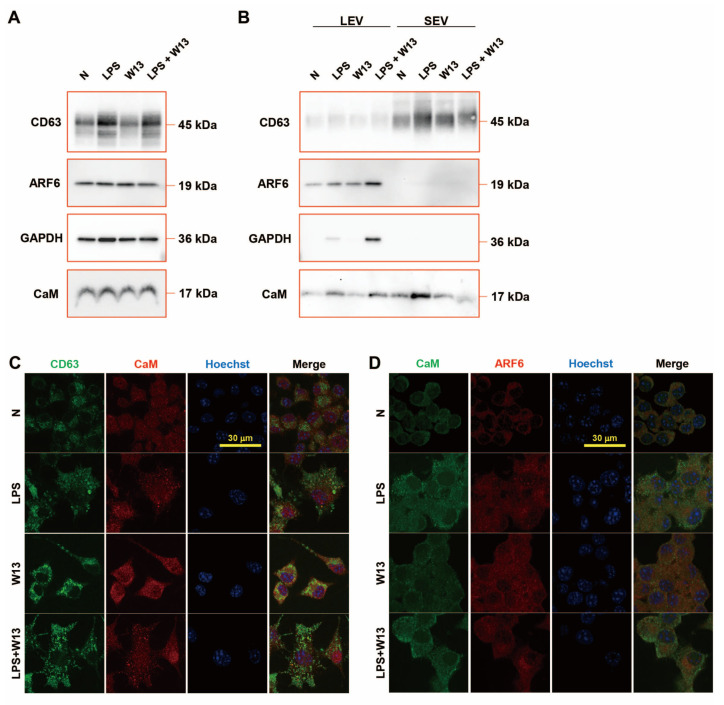
Extracellular vesicle-related proteins in RAW-Blue cells and their extracellular vesicles in the presence of LPS and W13. (**A**) Western blot analysis of extracellular vesicle-related proteins in RAW-Blue cells in the presence of 100 ng/mL LPS and/or 10 μM W13. (**B**) Western blot analysis of extracellular vesicle-related proteins in extracellular vesicles from RAW-Blue cells in the presence of LPS and/or W13. (**C**) Distribution of CD63 and CaM in RAW-Blue cells in the presence of LPS and/or W13. (**D**) Distribution of CaM and ARF6 in RAW-Blue cells in the presence of LPS and/or W13.

## Data Availability

The raw data can be provided by the corresponding author upon reasonable request.

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
