# Peer review of "Distribution of Signal Peptides in Microvesicles from Activated Macrophage Cells"

_ijms, 2023, doi:10.3390/ijms241512131_

Round 1
Reviewer 1 Report
This study analyzed the kinetics of signal peptide (SP) fragments in microvesicles (LEVs) and exosomes (SEVs) under inflammatory stimulations. Distribution of SP fragments is changeable and dependent on a presence of calmodulin (CaM). They concluded the analysis of SP and CaM in LEVs and SEVs is important for understanding cellular interactions and suggested the usefulness as biomarkers.
Comments as bellow:
1. The authors should make the lists of abbreviations, because many inconsistency and repeated explanations of abbreivations appeared.
2. The total size of manuscript is relatively large and some redundant looking. In “Result”, there are many interpretations, evaluations, and references. Is this the Journal’s style? These sentences should be moved to “Discussion”.
3. In human body, circulating EVs contain heterogenous groups of EVs originated from different cell-lineages. Please give some discussion on the mechanism(s) how specific this Macrophage-released EVs work target cell-specific in vivo.
4. There is a confusion about “definition of cargo molecules”. CD63 is a membrane protein, is not a cargo protein. How about ARF6, GAPDH, Cam. The author should give the clear explanation of localization and roles of each protein.
5. Lines 415 to 433: Many sentences depend on interpretation of previous studies. The author should concentrate to interpret their own data.
6. Line 25: What is the meaning of “orientation”? In the end of the same sentence, a word: “transportation” appears. If these two words are equal, I think “transportation” is more popular and is easily understandable. The same thing in Line 274-275, and in everywhere.
7. Lines 56 to 70. This part for explanation on biology of SP and its fragments may be better to have some illustration for promoting clear understanding.
8. This is an in vitro culture system containing only RAW-Blue cells. How about in vivo observation in infectious diseases. This explanation in “Discussion” would be helpful for readers to understand pathophysiology of infectious disease.
Minor comments:
1. Lines 113-118: Show the purity of each fraction of LEV and SEV.
2. Fig.1 C: LEV showed v-shape response to LPS stimulation. What is the explanation?
3. Fig.1, E: The band(s) of internal control should be shown in the same figure.
4. Fig.1 D: Lines 214-217, This protein analysis contains membrane protein as well as protein inside EV. How do you evaluate this data. Fig1D showed protein contents increased in parallel to conc. Of LPS. This means contents increased without changing size.
5. Fig.1 E no showing of internal control: GAPDH.
6. HRP in line 174 needs explanation.
7. Fig.1 A: n=3 means three at each concentration?
8. The word “m/z” suddenly appear at line 223 without explanation. The author should explain this word for readers in the other fields to get more visibility.
9. Lines 234: I do not understand what does mean “the orientation of SPs”. Change of distribution, or change the localization?
10. Line 257: There is no data on NF-kB, thus, “contribution by NF-kB” is hypothesis in this line. Sentence should be changed to that meaning.
11: Methods are WB and TF-MS. How the author differentiates membrane proteins and cytoplasmic protein. For example, CD63 is a surface protein on outer membrane of extracellular vesicles.
12. Line 273: The author should show the statics about difference between TNFa and IFNg, to make sure that the ratio of SEAP SP fragments per particle in SEV was higher in the presence of TNFa than in the presence of IFNg.
13. Lines 289-296. I do not understand what is happening in these experiments. Fig2B is showing that no effect of CaM blocking on the cell number, is it? (Line300)
14. If we think utilization of them as biomarkers, identification of cellular origin of LEVs and SEVs is important. It is better to give comments on this issue in “Discussion”.
15. Lines 361: The sentence makes no sense “We herein demonstrated for the first time that SP fragments were encapsulated in LEV and SEV, which is consistent with previous findings.” The author should make it clear which finding is their original and “for the first time”.
Writing was redundant, confusing, and immature. Intensive rewriting is required.
Reviewer 2 Report
Ono and cols. perform study where they demonstrate that a signal peptide from the human placental secreted alkaline phosphatase is present in extracellular vesicles released from activated macrophage cells. The work is interesting and novel. The experiments are rigorously conducted. However, they generalize in the title that signal peptides are distributed in microvesicles, but they demonstrate it only for one. The major question that arises for me is whether they think that this is a general phenomenon for signal peptides for all (or most of) the proteins or not; i.e. whether it is specific for SEAP or this phenomenon can be extrapolated to other proteins. In addition, extracellular vesicles are known to contain as cargo many different molecules, as well as debris from the cellular machinery activity, that can be delivered in a non-specific way. The question is: do the authors think that this signal peptide (and potentially others) are specifically directed to be packaged and secreted via extracellular vesicles, or is a random event? I.e. the SP is just “randomly” trapped into the vesicles lumen when they are formed. For this, it would be important to measure the concentration of the SP within the vesicles, and compare it with its concentration in other cell compartments (mainly the cytoplasm and/or the endoplasmic reticulum lumen).
Other minor comments:
Abstract:
-What do LEV and SEV correspond to? I mean, why do the authors abbreviate “microvesicles” as “LEV” and “exosomes” as “SEV”?
Materials and Methods:
-Lines 114 and 115, typo error: “centrifugated” —> “centrifuged”.
In general, the manuscript is very well written.
